# VEGF in Diabetic Retinopathy and Age-Related Macular Degeneration

**DOI:** 10.3390/ijms26114992

**Published:** 2025-05-22

**Authors:** Andrew Callan, Justin Heckman, Giani Tah, Samantha Lopez, Laura Valdez, Andrew Tsin

**Affiliations:** School of Medicine, The University of Texas Rio Grande Valley, Edinburg, TX 78539, USA; andrew.callan01@utrgv.edu (A.C.); justin.heckman01@utrgv.edu (J.H.); giani.tah01@utrgv.edu (G.T.); samantha.lopez15@utrgv.edu (S.L.); laura.valdez01@utrgv.edu (L.V.)

**Keywords:** diabetic retinopathy, age-related macular degeneration, VEGF, vascular endothelial growth factor, AMD, DR, Ophthalmology

## Abstract

Vascular endothelial growth factor (VEGF) plays a key role in angiogenesis throughout the human body, influencing countless physiological and pathological processes, including tumor growth, preeclampsia, and retinal diseases such as diabetic retinopathy (DR) and age-related macular degeneration (AMD). In DR, VEGF promotes retinal neovascularization and intraretinal fluid accumulation, leading to complications like diabetic macular edema (DME) and proliferative diabetic retinopathy (PDR). Regular intravitreal anti-VEGF injections are commonly used to manage PDR and DME, though repeated treatments are often required, and efficacy can be limited. AMD, a major cause of vision loss in older adults, is characterized by either dry or wet forms. While the dry form has not been shown to be influenced by VEGF, the choroidal neovascularization of wet AMD has strong associations with VEGF. Current treatment for wet AMD consists primarily of anti-VEGF injections, the gold standard of care, but is limited by varying patient responses, as treatments are often repeated every 4-8 weeks indefinitely. This review explores the pathogenic role of VEGF in both DR and AMD, discussing the molecular mechanisms underlying these diseases and the therapeutic approaches targeting VEGF. Despite advancements, the variability in treatment responses highlights the need for continued research to develop more effective therapies to prevent vision loss and blindness associated with these retinal diseases.

## 1. Introduction

Vascular endothelial growth factor (VEGF) is a critical growth factor that promotes angiogenesis throughout the human body. VEGF plays essential roles from embryonic development through adult physiological and pathological processes, including neovascularization in tumors, preeclampsia, and specific retinopathies [1]. The VEGF family includes placental growth factor (PlGF), endocrine gland-derived VEGF (EG-VEGF), VEGF-A, VEGF-B, VEGF-C, VEGF-D, VEGF-E, and VEGF-F [1,2,3]. All members share a cystine knot structure, formed by eight cysteine residues and three disulfide bonds, and function as either homodimers or heterodimers. These growth factors interact with extracellular tyrosine kinase receptors (VEGFRs), resulting in intracellular phosphorylation and activation of downstream angiogenic pathways [1,2]. VEGF receptors are found primarily on vascular and lymphatic endothelial cells, but are also expressed on macrophages, vascular smooth muscle cells, and renal mesangial cells. Of the main VEGFRs, VEGFR-1 mediates inflammatory cell migration, VEGFR-2 promotes endothelial migration and vascular permeability, and VEGFR-3 is involved in lymphangiogensis during tumorigenesis or inflammation [1]. VEGF’s central role in angiogenic signaling is crucial in the development of diabetic retinopathy (DR) and the wet form of age-related macular degeneration (AMD), both of which are characterized by pathological neovascularization at advanced stages [1,4].

Diabetic retinopathy (DR) is a frequent complication of long-standing diabetes mellitus (DM), affecting approximately 22% of people with diabetes globally [5]. Non-proliferative diabetic retinopathy (NPDR), the earlier stage, is marked by increased retinal vascular permeability, pericyte loss, microaneurysm formation, and acellular capillary development [4,5,6]. As the disease progresses to proliferative diabetic retinopathy (PDR), retinal neovascularization can lead to vitreous hemorrhage, tractional retinal detachment, and vision loss [4,7]. At any stage, increased vascular permeability and leakage may cause diabetic macular edema (DME), contributing to central vision loss [8]. Risk and progression of DR are primarily influenced by glycemic control, hypertension, dyslipidemia, sleep apnea, and genetic background [4,9]. Treatment options are based on severity and include anti-VEGF agents, corticosteroids, and laser photocoagulation for DME; anti-VEGF and laser photocoagulation for PDR; and vitreoretinal surgery for advanced disease [8,9]. Although anti-VEGF therapy can regress active neovascularization in PDR, it requires repeated administration and often does not halt progression in all patients [9]. Therefore, there is a need for more effective and durable treatments for DR [4].

Age-related macular degeneration (AMD) is a leading cause of irreversible central vision loss in older adults [10]. Major risk factors include aging, diet, smoking, cardiovascular disease, and genetic susceptibility [10,11]. Studies such as the Beaver Dam Eye Study show that AMD incidence increases after age 75 [12]. Vision loss in advanced AMD can occur via neovascular (“wet”) or non-neovascular (“dry”) mechanisms [13]. Dry AMD features macular drusen and dysfunction of the retinal pigment epithelium (RPE), the outermost retinal layer [14]. RPE cells are responsible for the recycling of visual pigment, the maintenance of photoreceptors, and the removal of waste products generated by cell metabolism [13,14]. RPE dysfunction leads to photoreceptor damage and drusen accumulation, resulting in impaired vision. Geographic atrophy (GA), seen in late dry AMD, is characterized by photoreceptor and RPE atrophy, and is the main vision-limiting feature of advanced dry AMD [14]. While there are no treatments that reverse GA, recent drugs have shown to slow progression via complement inhibition [15]. Neovascular (wet) AMD is marked by choroidal neovascularization, which leads to subretinal leakage, retinal detachment, fibrosis, and vision loss [13,16]. Anti-VEGF injections are the standard of care for neovascular AMD, but response varies due to patient and lesion-specific factors [16].

This review examines the current understandings of VEGF’s pathogenic roles in DR and AMD development and VEGF inhibition in the clinical treatment of these retinal diseases. A general overview of the role VEGF plays in the development of AMD and DR, which will be explored thoroughly in this paper, can be seen in Figure 1. Current knowledge of this topic is limited, and thus, further basic and clinical research will be needed to provide novel and effective DR and AMD treatments to protect vision and prevent blindness.

## 2. Pathophysiology of DR and the Role of VEGF

### 2.1. Non-Proliferative Diabetic Retinopathy NPDR and VEGF

VEGF is not the initiating factor in NPDR, but the retinal changes associated with NPDR promote increased VEGF expression, which predisposes to DME and progression to PDR [8,17]. The effects of long-standing DM causing upregulation of VEGF, are listed in Figure 1. Chronic hyperglycemia in DM induces inflammation, neurodegeneration, and microvascular damage in the retina [8,17]. The resulting hypoxia and ischemia promote acellular capillary formation and upregulation of VEGF [1,6,17]. Microscopically, acellular capillaries are generated by pericyte dropout and loss of endothelial cells due to the effects of chronic hyperglycemia on the retina [6]. Clinically, retinal microaneurysms are vascular dilatations in the retina and one of the earliest fundoscopic findings in DR [17,18]. These are a result of the lack of structural support caused by impaired oxygen delivery [18]. Retinal hypoperfusion is also associated with the development of blot hemorrhages and cotton wool spots, the latter of which are classically attributed to nerve fiber layer infarction, but axon rupture is not definitively supported as a mechanism by current literature [18].

DME is characterized by the onset of intraretinal and/or subretinal fluid in the macula of patients with either NPDR or PDR [8]. Fluid accumulation in the macula distorts the retina and impairs visual processing, leading to impaired visual acuity. The full mechanism for this has yet to be fully elucidated, but it has been shown that tissue hypoxia in the setting of NPDR and PDR elevates the production of VEGF [8]. Increased VEGF causes increased capillary permeability, resulting in the influx of fluid into the intraretinal and subretinal spaces. Other identified mechanisms contributing to fluid dysregulation in diabetic macular edema include Müller cell dysfunction [8]. Under physiological conditions, Müller cells facilitate the removal of fluid from the retina into the vitreous chamber or systemic circulation. Although Müller cell dysfunction has been implicated in diabetic retinopathy, it is currently unclear whether this disruption is a direct consequence of elevated VEGF levels or is mediated by other pathological processes associated with diabetic retinopathy [8].

### 2.2. Proliferative Diabetic Retinopathy PDR and VEGF

In response to ischemia and acellular capillaries established in NPDR, cells in the retina, including endothelial cells, RPE cells, and Müller cells, begin producing VEGF [1]. VEGF then acts as a signal to promote angiogenesis, the defining feature of progression from NPDR to PDR.

The transcription of VEGF is a process that is controlled by the presence or absence of oxygen its effects on hypoxia-inducible factors (HIF) [19,20]. The α-subunits of these transcription factors, including HIF1α and HIF2α, are hydroxylated in the cytoplasm in an oxygen-dependent manner by prolyl-hydroxylase domain proteins (PHDs). Of the known PHD isoforms, studies in mice have shown a predominance of PHD2 in the retina [21]. In normoxic conditions, PHDs hydroxylate these HIF1α and HIF2α with available molecular oxygen, which causes them to be ubiquitinated and degraded by the ubiquitin-proteasome protein degradation system [20]. However, in oxygen-poor conditions, the PHD enzymes are unable to hydroxylate HIF1α and HIF2α. This allows these transcription factor subunits to travel to the nucleus and dimerize with β-subunits, including HIF1β and HIF2β. This interaction results in the formation of an α,β-HIF heterodimer that is then able to interact with hypoxic response elements (HRE) on chromatin. This promotes the transcription of genes containing hypoxic response elements (HRE) in their promoter/enhancer regions, such as the VEGF-A gene. Increased expression of VEGF-A protein in hypoxic regions of the retina lead to retinal neovascularization of the inner retinal membrane, increasing vascular permeability, and contributing to the collapse of the BRB [20,22]. This vascular dysfunction is speculated to also be a byproduct of microglial cell activation and damage, as retinal microglia have been shown to respond to insult by releasing proteins like VEGF [23]. In PDR, VEGF plays a role as an inflammatory mediator that leads to leaky capillaries and neuronal damage, which in turn activates microglia to aggregate and create hyper-reflective intraretinal spots and worsening diabetic retinopathy [23]. As the disease progresses, the formation of fibrovascular membranes on the retinal surface due to chronic hemorrhaging severely distorts and occludes vision and can lead to tractional retinal detachment. [7]. Retinal detachment occurs through VEGF promoting angiogenic factors that stimulate neovascularization, leading to overall disruption of the basement membrane, resulting in vitreoretinal traction [24].

The optic nerve is also significantly affected by VEGF and is impacted by neovascularization and increased vascular permeability [25]. Microvasculature damage seen in diabetes and other systemic illnesses may cause decreased perfusion to the optic nerve, causing ischemic optic neuropathy [25]. In high-risk cases of PDR, neovascularization may be present on greater than one third of the area of the optic disk, with hemorrhages obscuring greater than the area of one optic disk [23].All in all, VEGF plays a significant role in proliferative diabetic retinopathy, as well as several other microvascular conditions of the retina, as it increases vascular permeability, creates fragile blood vessels, enhances inflammation, and alters blood flow, all leading to the progression of this disease and eventual permanent impairment of a person’s vision [26].

## 3. Pathophysiology of AMD and the Role of VEGF

### 3.1. Pathophysiology of Dry AMD and VEGF

VEGF has not been found to be involved in the pathogenesis of the dry stage of AMD [27]. However, hallmark findings in dry AMD, like drusen deposits and RPE dysfunction, are both a cause and a result of the change in perfusion and increase in inflammation occurring during this dry AMD [14]. The environment created in dry AMD precipitates an increase in VEGF production in order to improve perfusion in the choroid and outer retina, as a response to the hypoxia and dysregulation experienced in this highly metabolic tissue [14]. This relationship is presented in Figure 1.

### 3.2. Pathophysiology of Wet AMD and VEGF

The release of VEGF is known to mediate the abnormal angiogenesis seen in wet AMD, and long-term anti-VEGF therapy has remained the gold standard treatment to control choroidal neovascularization in patients with wet AMD for decades [27,28]. The pathogenesis of wet AMD begins during the dry stage, with an increase in VEGF expression and production [27]. The mechanisms by which VEGF is upregulated in dry AMD are central to understanding why progression to wet AMD occurs. Stress caused by drusen deposition beneath the RPE in dry AMD, as well as oxidative stress from aging or smoking promotes the conversion to wet AMD [27]. The resulting signal for angiogenesis with factors like VEGF results in the formation of choroidal neovascular membranes, but the exact mechanism for this process is believed to be multifactorial and not completely elucidated.

One pathway of increased VEGF production in the setting of dry AMD that is being explored involves the innate immune cells present and circulating through the retina [29]. The previously stated stressors to the perfusion and nutrition of the retina result in impairment of the lysosomal function of RPE cells leading to RPE cell death [17,30,31]. This environment of lysed RPE cell membranes and free intracellular proteins promotes the activation of retinal microglia to the “M1” state [29]. These M1 activated microglia have been shown to migrate to the subretinal space in the setting of chronic inflammation from dry AMD [29]. In the M1 state, microglia have shown to increase expression of injury responsive genes and decrease expression of homeostatic checkpoint genes like CX3CR1 [32]. These microglia are characterized by secreting interleukin (IL)-6, tumor necrosis factor (TNF)-a, IL-1b, and IL-12 and engaging in phagocytosis [33]. The production of these inflammatory mediators has been shown in mouse models to be strongly influence by the Akt2 pathway, which shows to be a promising target in the treatment of early, dry AMD [29]. The infiltration of activated M1 microglia and expression of these inflammatory mediators, therefore, activates subretinal neutrophils which have been shown to cause the RPE changes seen in AMD [29]. A summary of the current understanding of microglia activation pathways in AMD is depicted in Figure 2 [33]. The predominance of the M1 microglia phenotype over a long period of time contributes to the generation of GA, a defining feature of late-stage dry AMD with moderate to severe central vision impairment [33]. This microglia-neutrophil interaction resulting in pathologic tissue damage has also been found in other diseases like Alzheimer’s disease, intracerebral hemorrhage, and stroke [29].

As the “activation” stage progresses, microglia and macrophages shift to the M2a pathway, which produces IL-4 to inhibit the production of IL-6, TNF-a, IL-1b, and IL-12 [33]. This shifts the state from M1 to M2a and M2b, characterized by fibrosis and disciform scarring. M2b macrophages produce IL-10, TGF-β, and insulin growth factor (IGF)-1, and they function to induce angiogenesis and astrocyte scar formation. Microglia and macrophages can also polarize to the M2d phenotype, producing VEGF.in order to revascularize the wounded area [33]. Eventually, the setting shifts to the M2c state of “resolution”, where healthy tissue is tagged with a sialic acid signal for anti-inflammation, anti-phagocytosis, and anti-angiogenesis. The sialic acid is sensed by sialic acid-binding immunoglobulin-like lectin (SIGLEC) receptors on microglia, as well as by complement factor H [33]. It is of note that, as we age, sialic acid has been shown to decrease in cerebrospinal fluid [33]. The degenerated retina in late AMD has been shown to have decreased expression of molecules like sialic acid to signal healthy tissue, which is believed to be a factor contributing to the choroidal neovascularization seen in neovascular AMD [34]. When this state of resolution does not occur and macrophages are left in the M2a,b,d phase, cytokines like VEGF are constitutively produced, and angiogenesis continues unchecked [33]. In the clinical trials in the AREDS study, it was found that supplementation with zinc and other antioxidants helped reduce the risk of progression to advanced AMD in patients with extensive drusen, large drusen, or noncentral GA [35]. Zinc has been shown to polarize macrophages and microglia toward producing TNF-α, Il-6, and IL-8, the M1 pathway, and away from the VEGF producing M2d phenotype [36]. This effect is believed to be the reason zinc supplementation in vitamins like those studied in the AREDS trials help reduce the occurrence of choroidal neovascularization. It is important to note that while reducing the polarization to the M2a,b,d pathway helps prevent wet AMD, it does not show benefit for the prevention of GA [33].

The complement cascade has also been implicated in the pathogenesis of AMD and its progression to neovascularization [37]. C3a and C5a have been shown to be present in RPE cells 4 h after laser-induced retinal injury in mice models, and both C3a and C5a knockout mice showed to have a significant decrease in production of VEGF and recruitment of neutrophils [37]. The membrane attack complex (MAC), which is an assembly of complement meant to lyse a cell, has also been shown to induce VEGF production [38]. When the MAC is unable to complete cell lysis, the cell will secrete inflammatory mediators. In the case of RPE cells engaging with sublytic MAC, the RPE cells secrete VEGF, which contributes to the choroidal neovascularization seen in wet AMD [37]. A recent study in mice also demonstrated the function of complement in polarizing retina macrophages to the M2 state, which induces the angiogenesis seen in wet AMD [37]. Complement’s role in the AMD pathogenesis is currently being explored to see its potential use in treatment for AMD. This has been through the use of bispecific fusion proteins which target both VEGF and complement, like C3 and C5, to increase the anti-angiogenic effect of anti-VEGF treatment and prolong intervals between AMD treatments [37].

Among the possible combinations of VEGF ligands and receptors, the VEGF-A/VEGFR2 axis has been found to be the main pathway to promote angiogenesis in wet AMD [39]. A minor angiogenesis pathway includes VEGF-C or VEGF-D binding to either VEGFR2 or VEGFR3. The different VEGF isoforms have been found to be released by several different cell types in AMD, including retinal pigment epithelial (RPE) cells, Müller cells, and recruited macrophages [39].

In the setting of AMD, one mechanism of VEGF release involves immune cells migrating to the macula, where they are activated to secrete pro-angiogenic factors like VEGF-A, VEGF-C, and VEGF-D [39]. These ligands then bind to VEGFR2 and VEGFR3 on endothelial cells, promoting angiogenesis and vascular permeability. The resulting leakage damages photoreceptors and contributes to the irreparable vision loss involved in AMD progression [39]. Alterations in RPE cell-produced VEGF activity have also demonstrated involvement in AMD progression [40]. As Bruch’s membrane thickens in dry AMD, it limits the ability of VEGF to maintain choriocapillaries, which leads to the GA seen in late dry AMD. These results demonstrate the importance of VEGF activity to support a healthy blood supply to retinal tissues, suggesting healthy VEGF levels are necessary to support vision [40].

More recent studies have evaluated the role of VEGFR1 in angiogenesis [41]. It has been demonstrated in a study on PlGF that the interaction of PlGF and VEGFR1 promotes the recruitment of mononuclear phagocytes, creating an inflammatory environment that results in choroidal neovascularization [42]. Oppositely, VEGFR1 has also been shown to act as a decoy receptor for ligands like VEGF-A. This indirectly inhibits the pro-angiogenic activity of the VEGF-A/VEGFR2 axis [41]. Considering PlGF and VEGF-B also bind to VEGFR1, increased levels of these ligands can compete with VEGF-A, freeing up VEGF-A from being bound to VEGFR1. This promotes the interaction of VEGF-A with VEGFR2 and, therefore, angiogenesis. Because of this, it is critical to consider the effects of elevated levels of PlGF and VEGF-B in the setting of pathogenic neovascularization [41].

Since the advent of anti-VEGF inhibitors, which act exclusively on VEGF-A, an increased focus has now been given to VEGF-C and VEGF-D [43]. While VEGF-A has been identified as playing a key role in the promotion of choroidal neovascularization in wet AMD, a study done on patients receiving anti-VEGFA treatment demonstrated significant increase in levels of other angiogenic biomarkers including VEGF-C and angiopoietin-2 [44]. Similarly, an in vitro study on human RPE cells with blocked VEGF signaling showed an increase in IL-8 production [45]. These findings may explain the percentage of patients who show an inadequate response to VEGF-A inhibitors, demonstrating that VEGF-A is not the sole mediator of choroidal neovascularization in this disease process. A multifactorial investigation of the disease process will yield a more robust understanding and possibly open the door for more VEGF targets for future pharmacologic treatments [43].

## 4. Anti-VEGF Therapy for DR and AMD

Currently, there are a variety of therapeutics that have been developed for the treatment of both DR and AMD, listed in Figure 3, with innovations in therapeutics being introduced every year [46,47]. Figure 3 lists the current major treatment methods for both AMD and DR. In DR, common treatment options include intravitreal anti-VEGF injections, pan retinal photocoagulation, and vitrectomy [46]. AMD treatment also frequently consists of anti-VEGF, and additional options like photodynamic therapy may be included when indicated [47]. However, anti-VEGF therapies remain one of the top therapeutic choices for disease management due to their efficacy at targeting blood vessel angiogenesis and, therefore, slowing disease progression. The main mechanisms of anti-VEGF therapy currently used in clinical settings include drugs that inhibit the binding between VEGF and VEGF receptor (VEGFR) and decrease the downstream effects of this interaction. This results in a decrease of endothelial cell proliferation, vessel angiogenesis, and vascular permeability [48]. These therapeutics are currently available through intravitreal injections which allow the therapeutic effects to remain localized to the eye and prevent secondary reactions throughout the rest of the body [49]. There are several VEGF inhibitor medications available for the treatment of DR and AMD, and the most commonly used include aflibercept (brand name Eylea), bevacizumab (brand name Avastin), and ranibizumab (brand name Lucentis) [49].

Aflibercept is a 115kDa recombinant fusion protein composed of the Fc segment IG1 and found in Domain 2 of VEGF-R1 and Domain 3 of VEGF-R2 [50,51]. This competitively binds VEGF-A and VEGF-B with a higher affinity than their native receptors and differentiates Aflibercept from the other first-line anti-VEGF therapies, which are only able to bind VEGF-A. Aflibercept may be salvaged from protein catabolism due to the recycling of the Fc region to the Fc receptor. The intravitreal half-life is estimated at 7.1 days after injection, but clinical activity of aflibercept is seen for up to 2.5 months after injection [50]. Aflibercept has shown to bind even tighter to all VEGF isoforms compared to native VEGFRs, and it has almost a 100-times greater binding affinity to VEGF as compared to similar anti-VEGF injections like bevacizumab and ranibizumab [50]. Bevacizumab and ranibizumab are both human recombinant VEGF monoclonal antibodies [52]. Bevacizumab is a 149 kDa full-length antibody with an Fc region, contributing to receptor recycling and its approximately 20-day half-life. It was the first drug approved by the US Food and Drug Administration (FDA) for tumor anti-angiogenesis, whereas ranibizumab is a second-generation drug with a higher affinity for VEGF-A than the former. Ranibizumab lacks the Fc region and is, therefore, metabolized much faster in the body and has an approximate half-life of 2–3 h [52]. Approved by the FDA for the treatment of wet AMD in 2022, faricimab (brand name Vabysmo) is a newer anti-VEGF drug that utilizes a secondary Fab region to also bind to Angiopoitein-2 (Ang-2) [53]. By including the Ang-2 binding region on the anti-VEGF IgG monoclonal antibody and, therefore, improving vascular stability and VEGF-A desensitization, as demonstrated in the TENAYA and LUCERNE trials, faricimab has been shown to provide visual benefit to patients with up to 16-week treatment intervals [53,54]. Since its approval for wet AMD, faricimab has now been approved for diabetic macular edema and edema secondary to branch retinal vein occlusion [54].

Following the diagnosis of DR or AMD, a patient can be started on an anti-VEGF medication and be placed on a specific dosing regimen that the patient and physician agree on, typically beginning at a treatment interval of every 4 weeks [55,56]. Due to cost, issues in real-life adaptation, and risks of side effects that come with a fixed dosing schedule, anti-VEGF medications can be offered in two different regimens including “pro re nata” (PRN) and “treat and extend” (T&E) which are variations from the original fixed dose regimen followed in clinical trials [57]. PRN is a regimen where treatment occurs as needed while patients continue to follow up in a fixed period to monitor the gradual changes in optical coherence tomography (OCT) scans and visual acuity between visits and injections. The T&E modality begins with a fixed dosing regimen and gradually increases the time between doses when a patient obtains a stable condition. Each modality has varying benefits depending on the disease and drug chosen. Once a patient has been started on treatment, each follow-up visit will consist of examining and tracking the efficacy of treatment, as mentioned previously, through visual acuity tests and assessments of the morphology of the macula on OCT. On OCT for AMD progression, physicians normally look at intraretinal fluid, subretinal fluid, central retinal thickness, and intraretinal cysts [16]. In DR, OCT is performed to monitor retinal thickness, progression of macular edema, and microaneurysms [55]. Additionally, fundus photography can be utilized in DR for neovascularization. Progression and changes in treatment from this point on will highly depend on present illness and patient response to treatment [55].

### 4.1. Anti-VEGF Therapy for NPDR

The main objective of anti-VEGF treatments in NPDR has been to target and slow down disease progression to PDR [58]. Currently, anti-VEGF therapies are only indicated for moderate to severe NPDR and are not recommended for use in mild stages unless the patient has been diagnosed with DME, in which case anti-VEGF therapy should be considered. Despite this, it has been shown that starting anti-VEGF treatment without the presence of DME still had positive effects on managing disease progression [58]. Previous studies, specifically DRCR Protocol W, have shown that patients undergoing NPDR treatment minus DME with aflibercept had a 16.3% probability of CI-DME onset or PDR progression vs a 43.5% probability in the control group at the two years mark out of their four years of study [59]. It is important to note, however, that administering anti-VEGF therapeutics at this stage, despite their demonstrated benefit, can still present with unfavorable situations such as increased cost, over-treatment when not necessary, and increased side effect risks [60]. Currently, ranibizumab and aflibercept are both approved for use in DR of any level, with or without DME being present. In addition to ranibizumab and aflibercept, bevacizumab is commonly used as an “off label” therapeutic for NPDR as well. Between the three medications, studies have shown that in NPDR, there is no significant difference in halting disease progression from NPDR to PDR, and they all improve retinopathy with similar results [60].

### 4.2. Anti-VEGF Therapy for PDR

Compared to NPDR, anti-VEGF treatment for PDR, with and without DME, is a mainstay treatment modality. In comparison to PRP, anti-VEGF has been shown to have an increase in mean visual acuity letter score, with some studies showing a +2.8 letter increase with anti-VEGF treatment in comparison to a +0.2 increase for PRP [61]. Furthermore, other benefits in comparison to PRP included a decrease in peripheral vision loss, reduced treatment failure, and decreased retinal detachment. Similarly to NPDR, the main anti-VEGF drugs available for PDR include aflibercept, ranibizumab, and bevacizumab [55]. The main objective with treatment at any stage in PDR is to prevent PDR progression, which is measured through DRSS scores, non-perfusion index, and leakage index. Currently, the DR FDA recommendations for aflibercept are 2 mg monthly for the first five months of treatment [62]. Following this, clinicians are advised to continue 2 mg dosing every two months. For ranibizumab, the recommended dosing is 0.3 mg once a month. For bevacizumab, since it is not FDA approved for DR use, there are no specific recommendations; however, studies have used 1.25 or 2.5 mg as standard dosing [63,64].

As anti-VEGF began to gain popularity and slowly move toward being the main treatment suggested to patients, many studies have investigated combining treatments and showing how anti-VEGF, when used in conjunction with PRP or vitrectomy, can alter patient outcomes. Anti-VEGF therapy, when used in conjunction with PRP, for example, had an improved BCVA compared to that of PRP alone [65,66]. A similar result was found in patients receiving vitrectomy plus anti-VEGF with patients achieving an improvement in BVCA by 6 months after the beginning of treatment [67].

### 4.3. Anti-VEGF Therapy for Dry AMD

Not only is dry AMD not treated with anti-VEGF, but studies have begun to show slight speculations that the use of anti-VEGF treatment in wet AMD has shown a further progression of dry AMD [68]. In addition to side effects and complications of anti-VEGF medications, such as endophthalmitis, it has been noticed that patients with dry AMD who received consistent anti-VEGF injections had a slight correlation to increased incidence of GA [68]. As mentioned previously, GA presents in late-stage dry AMD and is characterized by atrophic lesions that progress centrally in the retina. This association, however, is not definite, and more studies need to be conducted to identify a strong incidence pattern. Other than this, there is no other correlation or association of anti-VEGF medication with dry AMD [68].

### 4.4. Anti-VEGF Therapy for Wet AMD

Currently, anti-VEGF intravitreal injections remain the first-line therapeutic for wet AMD. Similarly to DR, wet AMD can be treated with the same anti-VEGF compounds while including a few more that are currently not indicated for use in DR, such as brolucizumab and faricimab [69,70]. In a two-year trial comparing the efficacy of ranibizumab to bevacizumab, there was no significant difference between the mean visual gain when treated using the same regimen [71]. Furthermore, studies comparing ranibizumab and aflibercept have also shown similar results with both effectively acting on wet AMD [72]. As a result of the similarity in their therapeutic ability, patient preference and circumstances take precedence over which treatment the physician chooses to prescribe. Out of all five, bevacizumab, which is used “off label”, is more commonly prescribed than the others due to its cost-effectiveness and high rates of insurance approval [73].

Similarly to DR, the length of treatment and number of injections a patient receives per year depend on the physician-prescribed dosing regimen [56]. For wet AMD, there are both PRN and T&E in which the latter has been shown to be more effective in increasing visual acuity. T&E modalities, in both ranibizumab and aflibercept, have been shown to be more effective than their PRN counterparts at 24 months with a mean visual acuity of 58.9 compared to 62.2 [56,72]. Currently, the FDA recommends a 2 mg dose regimen for aflibercept for the first three months followed by the same dosage every 2 months [62]. This treatment could be extended to a 2 mg dose every 12 weeks; however, patients should be monitored constantly for changes [62]. For ranibizumab, there is slightly more freedom with dosing regimen in comparison to DR with the first being a monthly dose of 0.5 mg every month. Other options that are FDA approved are starting off with three monthly doses and then reducing the frequency depending on patient presentation during evaluation [74]. Additionally, patients can receive monthly doses for the first 4 months and after that, it can be given every 3 months [74].

A concern with using T&E is the need for more frequent injections, which could potentially lead to increased side effects or complications such as endophthalmitis, intraocular pressure elevation, and intraocular inflammation [75]. Therefore, drugs with a longer duration of action are desired to limit side effects and extend treatment intervals. This has led to medications such as faricimab, to be developed. Faricimab, a recent FDA-approved VEGF and Ang-2 inhibitor, has been shown to work as effectively as other anti-VEGF drugs in treating wet AMD, but it does not require as many injections [50,51]. With the use of faricimab, treatments could be increased to quarterly instead of bimonthly aflibercept or ranibizumab. In addition to established better patient compliance with decreased injections needed, it can help reduce the risk of complications associated with frequent injections [50,51].

## 5. Conclusions

VEGF is a critical component of the pathogenesis and progression of DR and AMD. VEGF promotes angiogenesis and increases overall vascular permeability, contributing to the development and exacerbation of these retinal diseases. In DR, VEGF is a key mediator that facilitates the proliferation of retinal capillaries in PDR. It creates a ripe environment for neovascularization, vitreous hemorrhage, optic nerve damage, and ultimately total vision loss. In contrast, VEGF facilitates wet AMD through choroidal neovascularization, resulting in subretinal hemorrhage, fibrosis, and severe central vision impairment. The major axis for this occurrence is the VEGF2/VEGFR1, which is the major target of our current anti-VEGF medications. However, the impact of other VEGF ligands and receptors is being studied, as it is the presence of these less-studied pathways that may contribute to treatment resistance. Though targeting the VEGF2/VEGFR1 pathway has been the standard of care, that standard may change as research into other treatment strategies develop. The introduction of anti-VEGF therapies has greatly improved the management and slowed the progression of PDR and Wet AMD. Although these treatments have demonstrated efficacy in preventing vision loss and, in some cases, improving visual acuity, challenges remain, including variation in patient reaction and the need for retreatment, such as frequent intravitreal injections. At this time, equally effective alternative treatments to anti-VEGF injections have not been identified. It is essential to conduct additional research to mitigate VEGF, develop new therapeutic targets, and enhance the understanding of the molecular mechanisms underlying the role of VEGF in both DR and AMD.

## Figures and Tables

**Figure 1 ijms-26-04992-f001:**
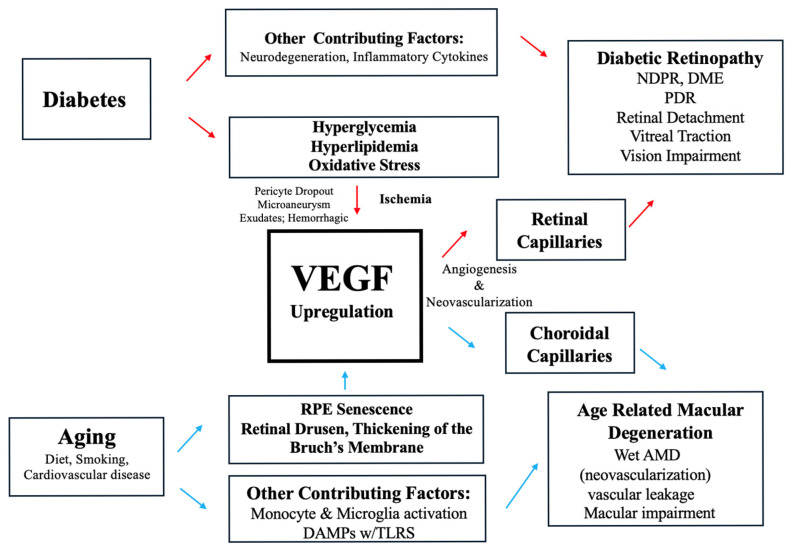
The central role VEGF plays in the development of DR and AMD. Blue arrows are assigned to the AMD pathway, and red arrows are assigned to the DR pathway.

**Figure 2 ijms-26-04992-f002:**
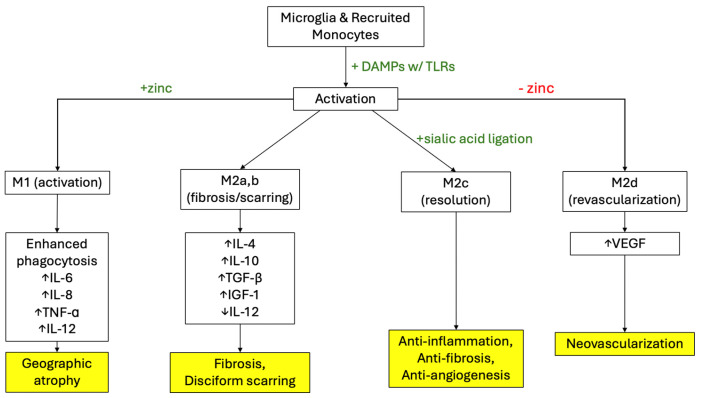
The role of the major pathways of retinal monocytes and microglia activation and polarization in AMD development. Mediators of these pathways include zinc and sialic acid ligation. + denotes a positive influence on that pathway. - denotes a negative influence on that pathway.

**Figure 3 ijms-26-04992-f003:**
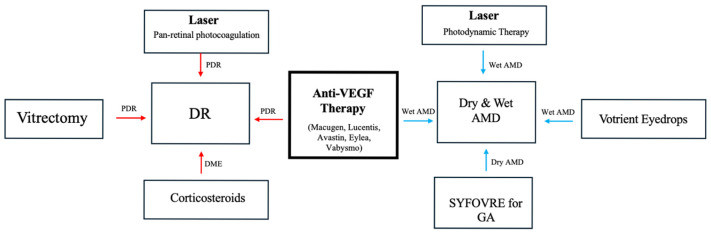
Anti-VEGF therapy in wet AMD and DR. Blue arrows are assigned to the AMD pathway, and red arrows are assigned to the DR pathway.

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
