# Peer review of "VEGF in Diabetic Retinopathy and Age-Related Macular Degeneration"

_ijms, 2025, doi:10.3390/ijms26114992_

Round 1

Reviewer 1 Report

Comments and Suggestions for Authors

This is a good review on the role of VEGF on diabetic retinopathy and age-related macular degeneration, it is mainly focused on angiogenesis and its treatment.

I have some comments:

  1. I found the work somehow repetitive, particularly lines 120-122 and 138-140 and section 1B.
  2. Line 154-155, “ In response to ischemia and acellular capillaries established in NPDR, cells in the retina, including endothelial cells, RPE cells, and Müller cells, begin producing VEGF. This is described in Figure 1”. It is not, in fact the figure (s) is  (are) general, not explains any mechanism.
  3. Legend figure 1,” Red arrows are assigned to the AMD pathway, and blue arrows are assigned to the DR pathway”. In the figure the colors are opposite; the same happens in the figure 3.
  4. Lines 127-128,” This results in direct interference of cell adhesion molecules necessary for the 127

blood retinal barrier”. That is probably a mistake, although adhesion molecules are important, the tight junctions molecules maintain the blood retinal barrier.

  1. Line 221-223. Reference 28 is not related.
  2. Please indicate the meaning of all abbreviations, in order to better follow the study.
  3. As the authors mentioned there are different anti VEGF molecules and treatments, according to the patients and medical doctor; then, it would be interesting if the authors could discuss on possible standardization, and alternative treatments.

Author Response

We would like to thank you for your review of our manuscript. We have diligently addressed your comments and have detailed what changes we have made here.

  1. I found the work somehow repetitive, particularly lines 120-122 and 138-140 and section 1B.
    - Lines 138-140 have been removed to eliminate redundancy.

2. Line 154-155, “ In response to ischemia and acellular capillaries established in NPDR, cells in the retina, including endothelial cells, RPE cells, and Müller cells, begin producing VEGF. This is described in Figure 1”. It is not, in fact the figure (s) is (are) general, not explains any mechanism.
- This section has been revised to no longer reference Figure 1, as the figure does not illustrate this mechanism.

3. Legend figure 1,” Red arrows are assigned to the AMD pathway, and blue arrows are assigned to the DR pathway”. In the figure the colors are opposite; the same happens in the figure 3.
- All figure captions have been adjusted to correctly identify the color to the diagnosis.

4. Lines 127-128,” This results in direct interference of cell adhesion molecules necessary for the 127 blood retinal barrier”. That is probably a mistake, although adhesion molecules are important, the tight junctions molecules maintain the blood retinal barrier.
- This mistake has been corrected. This section has been revised to no longer refer to cell adhesion molecules, and rather discuss the topic of blood-retinal barrier disruption correctly.

5. Line 221-223. Reference 28 is not related.
- Reference 28 has been removed from this section, as that paragraph should reference #29 completely.

6. Please indicate the meaning of all abbreviations, in order to better follow the study.
- All abbreviations throughout the manuscript have been reviewed and corrected.

7. As the authors mentioned there are different anti VEGF molecules and treatments, according to the patients and medical doctor; then, it would be interesting if the authors could discuss on possible standardization, and alternative treatments.
- This is now briefly addressed in the conclusion paragraph. Standardization of Anti-VEGF therapy is not a well-written about topic because doctors tend take personal preference over what drugs they use based on data they're read and personal experience, and with the expansion of the drugs available, it has become less and less likely that a standardization of the treatment of these conditions will predominate. There is no alternative to anti-VEGF use besides the previously described laser therapies describes in the beginning of Section 4.

We greatly appreciate the thorough review of our manuscript and look forward to further review.

Reviewer 2 Report

Comments and Suggestions for Authors

The review by Callan A et al. focused on VEGF's role in diabetic retinopathy and age-related macular degeneration pathologies. To contrast its effect, therapies are available based on the neutralization of the molecule, such as anti-VEGF compounds, which are administered primarily through intraocular injections. Even if these kinds of treatment are golden standards for the angiogenic/neovascular-related diseases, the research and the therapeutic approaches are continuously evolving to improve patient quality of life. Even though there are reviews on the role of VEGF, the one by Callan A et al. could be useful to the researchers of the field to have good and easily understandable information related to VEGF and therapeutics, updated with recent literature. So, the review can be published in the present form.

Author Response

We greatly appreciate your thorough review of our manuscript.